# Validity and Reliability of Polar Team Pro and Playermaker for Estimating Running Distance and Speed in Indoor and Outdoor Conditions

**DOI:** 10.3390/s23198251

**Published:** 2023-10-05

**Authors:** Simen Sandmæl, Roland van den Tillaar, Terje Dalen

**Affiliations:** Department of Sports Sciences, Nord University, 7600 Levanger, Norway; roland.v.tillaar@nord.no (R.v.d.T.); terje.dalen@nord.no (T.D.)

**Keywords:** motion analysis, athlete tracking, athlete monitoring, sports technology, global positioning system, inertial measurement unit

## Abstract

Although global positioning systems and inertial measurement unit systems are often used to quantify physical variables in training, both types of systems need to be compared, considering their frequent use in measuring physical loads. Thus, the purpose of our study was to test the reliability and validity of speed and distance run measurements at different intensities in indoor and outdoor conditions made by Polar Team Pro and Playermaker. Four participants (age = 30.0 ± 5.1 years, body mass = 76.3 ± 5.3 kg, height = 1.79 ± 0.09 m), each wearing three Polar Team Pro and two Playermaker sensors, performed 100 m runs with different prescribed intensities (i.e., criterion measure) varying from 8 to 24 km h^−1^, in a straight line and/or rectangle under indoor and outdoor conditions. Both systems underestimated total distance; Playermaker underestimated speed, the extent of which increased as speed increased, while Polar Team Pro overestimated mean speed at 8 km h^−1^ for the straight-line condition. No differences emerged in mean speed estimated by Polar Team Pro at any intensities other than 20 km h^−1^, which was underestimated by 2%. The reliability of the sensors was good, given a coefficient of variation (CV) of <2% for all conditions except when measuring indoor conditions with Polar Team Pro (CV ≈ 10%). Intraclass correlations (ICCs) for consistency within the sensors varied from 0.47 to 0.99, and significantly lower ICCs were documented at 8, 10, and 12 km h^−1^. Both systems underestimated distance measured in indoor and outdoor conditions, and distance validity in different intensities seemed to worsen as speeds increased. Although Polar Team Pro demonstrated poor validity and reliability in indoor conditions, both systems exhibited good reliability between their sensors in outdoor conditions, whereas the reliability within their sensors varied with different speeds.

## 1. Introduction

Monitoring external loads that athletes confront during regular training sessions is important for achieving the desired outcomes of training [1]. Enhancing performance requires a relatively high training load combined with a recovery phase; however, a training load that is too high or an inadequate recovery phase can compromise athletes’ likelihood of remaining free of injury [2]. Of the several means of controlling an athlete’s external training load during training sessions, one is monitoring distances and total distance at different intensities. In many sports (e.g., track-and-field running), the external load can be easily controlled by instructing athletes to run prescribed distances at different intensities [3]. In team sports such as soccer, however, multiple factors affect speed during training that make controlling speed relatively difficult. Therefore, in the past two decades, sport-tracking system suppliers (e.g., GPSports, Catapult, and STATSports) have increasingly developed devices capable of quantifying external training loads using microtechnology, such as global positioning systems (GPS) and/or inertial measurement unit devices (IMUs) [4].

On the one hand, GPS technology has considerably improved in the past decade, largely due to increasingly high sampling rates, but on the other hand remains satellite-based and therefore restricted to outdoor use [5]. Several studies have shown the good validity of total distance measurements using GPS regardless of the GPS device’s sampling rate (i.e., 1, 5, 10, 15, or 18 Hz) [6,7,8,9,10,11]. By contrast, the validity of speed measurements seems to be limited and to depend on sampling rates, such that higher sampling rates provide greater validity [9]. Using 10 Hz GPS has allowed measuring accurate, precise values for distances in different speed zones but shown slightly greater variation with higher speeds [12]. Even so, using 10 Hz GPS has also provided distance measurements with acceptable validity during short sprints (e.g., 15 m and 30 m) [13], specific movements in field-based team sports activities [14,15], and high-speed running in field-based team sports [11]. The validity of top-speed measurements during field-based activities has been acceptable as well [7,15]. Overall, the consensus in the literature is that the validity of using 10 Hz GPS during field-based activities decreases as athletes’ speed increases [6,11,14]. Along those lines, using 10 Hz GPS has shown a high degree of validity in measuring instantaneous speed within constant running speeds and for running that involves accelerations, the validity of which increases proportionally with higher initial speeds [12]. Moreover, though studies have shown that GPS devices typically measure below mean speed and the total distance measurement criterion, such results have been inconsistent [15,16]. Regardless of the device’s sampling rate, multiple studies have revealed the good reliability of GPS devices in measuring total distance [11,12,13,17], and their findings suggest that a device’s higher sampling rate increases the reliability of speed measurements [9,17]. Although the sampling of GPS devices at 10 Hz has been tested in different speed zones and revealed acceptable reliability [5,18], the inter-unit reliability of the devices has been poor, with a coefficient of variation (CV) ranging between 20% and 78% when comparing devices of the same model from the same manufacturer [8,19].

On the other hand, IMUs have increasingly been used as an alternative to GPS to measure speed-based metrics in team sports, largely because IMUs do not depend on satellite signals. IMUs are capable of assessing an athlete’s stride length, frequency, and running pattern [20,21], and can be fitted to track speed-based variables and thereby monitor physical variables in team sports [22]. By measuring an athlete’s accelerations and angular speed, changes in gait can be detected for each stride, and the speed and orientation of various body parts can be determined [23,24]. Beyond that, a comparison of foot-worn IMUs and GPS devices revealed that IMUs recorded higher speeds and greater distances than GPS, though no differences emerged in peak speed between the two types of devices. Mean speeds measured using IMUs were also significantly higher than speeds measured with 10 Hz GPS devices, and inter-unit reliability between foot-worn IMUs was found to be satisfactory [25] Added to that, IMUs have broader applicability because they can be used both indoors and outdoors [22].

Although sport-tracking systems are relatively expensive and thus not available to every sports team, the Polar Team Pro and Playermaker systems, which include GPS and IMU, are inexpensive. Added to that, they claim to accurately measure distances at different intensities in both indoor and outdoor situations; however, evidence to (dis)confirm those claims remains absent in the literature. Thus, the aim of our study was to test and compare the reliability and validity of speed and distance measurements made by Polar Team Pro and Playermaker at different intensities in indoor and outdoor conditions.

## 2. Materials and Methods

A repeated measures design was implemented to test the reliability and validity of Polar Team Pro and Playermaker during runs of different intensities outdoors in straight lines and both indoors and outdoors in rectangles performed over the course of several test sessions. The straight-line and 30 m × 20 m rectangular running conditions involved running a distance of 100 m; the total number of 100 m runs performed in each test session appears in Table 1. Total distance, distance, and speed at different running intensities were used as variables and compared to control-measured 100 m distances. Reliability was tested in terms of systematic bias, random error, and test–retest correlation [26,27].

### 2.1. Participants

Four participants (30.0 ± 5.1 years old, 76.3 ± 5.3 kg body mass, 1.79 ± 0.09 m body height) participated in the tests. All participants provided their informed consent to participate prior to testing, and the tests were approved by the Norwegian Centre for Research Data and conformed to the latest revision of the Declaration of Helsinki.

### 2.2. Procedure

After 5 min of individualized warm-ups consisting of jogging and dynamic stretching, the participants were instructed to stand beside one another on the starting line. A test leader gave audible signals to indicate both when the participants should begin jogging at a prescribed pace and at every 25 m to ensure that they maintained the set pace of 8 km h^−1^ in both the straight-line and rectangular running conditions. Continuing at that pace, participants in the straight-line condition reversed direction after 100 m and, in the 30 m × 20 m rectangle, changed direction at every corner. After the 400 m run at 8 km h^−1^, the pace was increased to 10 km h^−1^ for two 200 m runs, followed by 48 s of rest after the second run. After that, three 100 m runs were performed at a pace of 12 km h^−1^, with 30 s of rest after the third run, followed by two 100 m runs at 15, 18, and 20 km h^−1^ in all conditions and at 24 km h^−1^ in the straight-line condition. Participants rested after every 100 m run starting from 15 km h^−1^ until 1 min had elapsed. Table 1 details the running speeds, running times, and periods of rest between the runs. While resting, participants were required to stand still to prevent the systems from detecting additional meters moved. Altogether, 1700 m was prescribed in the straight-line running condition and 1500 m in the indoor and outdoor rectangular running conditions.

In the straight-line condition, a cone marked the distance every 25 m to ensure that all participants ran at the prescribed pace. By contrast, the 20 m × 30 m rectangle was marked with cones every 5 m, and every participant started at their own cone behind the other participants in order to maintain the same distance between themselves while running. One participant with experience in maintaining a prescribed pace served as the pacer; in the straight-line condition, the participants continued to run the same line, whereas the pacer in the rectangular running condition led the entire group (see Figure 1).

Each participant wore three Polar Team Pro belts (Polar Team Pro, Polar Electro, Kempele, Finland) around their chest and two sets of foot-mounted IMUs (IMU, Playermaker, Tel Aviv, Israel) on their feet. The IMUs consisted of a tri-axial accelerometer and a tri-axial gyroscope (MPU-9150, InvenSense, Sunnyvale, CA, USA) used to measure speed and rapid changes in the speed of each foot during gait, respectively. Both systems were used according to the described procedure for each. Polar Team Pro, which was used in the indoor conditions, consisted of a GPS system and an IMU. Positional data were sampled at 10 Hz for both systems. Once raw data (i.e., time, speed, and distance) were exported to an Excel spreadsheet for further analysis, the total distance for every participant in each condition was calculated and compared with the actual total distance. Raw data from each 100 m run were also calculated based upon the prescribed time for each run and the mean speed for the distance in each condition. All calculated values were subsequently subjected to statistical analysis.

### 2.3. Statistical Analysis

Validity and reliability were tested in terms of systematic bias, random error, and test–retest correlation, and the Shapiro–Wilk test was used to verify the normality of all variables. Systematic bias was calculated for the entire training session according to speed category and for each 100 m run between prescribed distance, speed, and measured parameters, after which it was calculated using a 3 (condition: straight line, outdoor rectangle, and indoor rectangle) × 2 (device: Polar Team Pro and Playermaker) × 6 (speed: 8–20 km h^−1^) analysis of variance (ANOVA) with repeated measures. When a significant effect was found, two- and one-way ANOVAs were performed on each factor to identify where differences had emerged, and when the sphericity assumption was violated, Greenhouse–Geisser adjustments of the *p* values were recorded. Effect size was evaluated with partial η^2^, such that 0.01 < η^2^ < 0.06 constituted a small effect, 0.06 < η^2^ < 0.14 a medium effect, and η^2^ > 0.14 a large effect [28].

Random error between the different sensors of each measuring system was estimated using a coefficient of variance (CV) as the standard deviation between the sensors divided by the mean of the sensors and multiplied by 100 for measured speed and distance. Different described speeds in which the CV was less than 10% were considered to be good [28]. To evaluate test–retest reliability, two to four runs were calculated at each speed using the intraclass correlation (ICC) for consistency and the CV for each test condition and device. A 3 (condition) × 2 (device) × (speed) repeated measures ANOVA was performed to compare the CV and ICC reliability of the three conditions, intensities, and devices. As for the thresholds for interpreting ICC results, 0.20–0.49 was low, 0.50–0.74 was moderate, 0.75–0.89 was high, 0.90–0.98 was very high, and ≥0.99 was extremely high. A mean within-session reliability with an ICC ≥ 0.67 and a CV ≤ 10% was considered to be acceptable; with an ICC < 0.67 and CV > 10% to be moderate; and with an ICC < 0.67 and a CV > 10% to be poor [28,29,30]. The level of significance was set at *p* ≤ 0.05, and all data were recorded as M ± SD. Statistical analysis was performed with IBM^®^ SPSS^®^ Statistics 24.0 (SPSS, Inc., Chicago, IL, USA).

## 3. Results

Total running distances measured with Polar Team Pro and Playermaker were underestimated in every condition, and measurement conditions exerted a significant effect (*F* = 31, *p* < 0.001, η_p_^2^ = 0.81) on the percentage difference with actual distance. Although no significant effect emerged for either measuring device (*F* = 2.3, *p* = 0.174, η_p_^2^ = 0.24), a significant condition-to-device interaction did emerge (*F* = 10.9, *p* = 0.012, η_p_^2^ = 0.61). A post hoc comparison revealed an increased percentage difference from the straight-line condition to the outdoor rectangle condition to the indoor rectangle condition for both devices. In the straight-line condition, Polar Team Pro presented a significantly lower measured percentage difference than Playermaker, whereas the opposite occurred in the indoor rectangle condition (see Table 2).

Concerning the prescribed speed measurements, significant effects emerged related to running condition, speed, measuring device, and all interactions (*F* ≥ 11.1, *p* < 0.01, η_p_^2^ ≥ 0.64). A post hoc comparison revealed that measured distances decreased as speeds increased; the measurements were closer to 100 m in the straight-line condition than in the outdoor rectangle condition, and the outdoor rectangle condition had more meters measured than the indoor condition. In general, Polar Team Pro measured longer distances than Playermaker in the straight-line condition, with no significant differences detected in the outdoor rectangle condition. However, in the indoor rectangle condition, Playermaker measured longer distances than Polar Team Pro (see Figure 2).

According to calculations of measured speed, Playermaker significantly underestimated mean speeds in all conditions at all speeds, which worsened with increased speed and varied between conditions (i.e., 2–12%). Meanwhile, Polar Team Pro significantly overestimated mean speeds at 8 km h^−1^ in the straight-line condition; however, no significant difference in estimated mean speed occurred with Polar Team Pro at other speeds, except at 20 km h^−1^, which was underestimated by 2%. Although variations in Polar Team Pro’s measurements in both rectangle conditions at mean speeds of 8 and 10 km h^−1^ did not differ significantly, increased mean speeds were significantly underestimated in both rectangle conditions (see Figure 3 and Table 3).

On average, the random error between the different sensors of Playermaker and Polar Team Pro was below 2% in all conditions and for the speed and distance parameters; however, such was not the case for the indoor rectangle condition with Polar Team Pro, which showed a random error of 9% (see Figure 4). Furthermore, the CV between sensors was always lower for Playermaker (i.e., 1.4%) than for Polar Team Pro (i.e., 2.0%). The CV between the sensors significantly increased from 8 and 10 km h^−1^ compared with the >12 km h^−1^ runs (*F*_14,154_ = 3.5, *p* < 0.001, η_p_^2^ = 0.24); however, the CV was significantly higher when measuring the indoor condition with both systems (*F*_2,28_ = 185, *p* < 0.001, η_p_^2^ = 0.94) than it was for the outdoor conditions, and was higher in the outdoor rectangle condition measured with Polar Team Pro than with the straight-line condition (i.e., 2.3% vs. 10.0%). No significant difference in the CV was detected when calculating distance or speed (*F*_1,14_ = 1.86, *p* = 0.194, η_p_^2^ = 0.11).

When the reliability within the sensors was tested according to the ICC and CV within each sensor at different intensities and conditions, the ICC varied from 0.471 (i.e., low) to 0.994 (i.e., very high), and significantly lower ICCs were found (*F*_5,55_ = 6.4, *p* < 0.001, η_p_^2^ = 0.37) at 8, 10, and 12 km h^−1^ than at the other three intensities. Moreover, the ICC significantly increased at 20 km h^−1^ compared with the other intensities when all factors were considered together (see Table 4). When the ICC was compared with the other factors, only a significant effect was found (*F*_2,5_ = 6.2, *p* = 0.015, η_p_^2^ = 0.55), and significant effect sizes were detected for the other factors and interaction effects (η_p_^2^ = 0.23–0.53) but not for the measuring device (η_p_^2^ = 0.03). A post hoc comparison revealed that the speed and distance ICCs in the indoor rectangle condition were significantly higher than in the other two conditions, and the ICC during the straight-line condition was significantly higher for Playermaker than for Polar Team Pro.

The test–retest CVs ranged from 0.29 to 6.39. A significant effect was only observed for speed (*F*_5,55_ = 3.78, *p* = 0.005, η_p_^2^ = 0.26) and measuring device (*F*_1,5_ = 12.4, *p* = 0.017, η_p_^2^ = 0.71). A post hoc comparison revealed that Polar Team Pro had a significantly higher CV in the straight-line and indoor rectangle conditions than Playermaker did (see Table 5).

## 4. Discussion

The aim of our study was to test and compare the validity and reliability of measurements made by Polar Team Pro and Playermaker of speed and distances run in different indoor and outdoor conditions. The major findings were that both systems underestimated measured distances and velocity in indoor and outdoor conditions alike and that the validity of distances in different intensities seemed to worsen as speed increased. Reliability appeared to be acceptable, except in indoor conditions, in which Polar Team Pro demonstrated poor validity as well as reliability.

Polar Team Pro measured significant lower distances and provided good validity in the straight-line outdoor condition (<1.1%). Despite the statistical significance, however, the marginal difference in distance does not confer practical significance. Likewise, albeit with a higher percentage of difference, Polar Team Pro measured significantly lower distances in the outdoor rectangle condition (i.e., <5%). Those findings are supported by the results of Huggins et al. [5], who reported good validity when assessing total distance measurements using Polar Team Pro (i.e., 10 Hz GPS). For the indoor rectangle condition, Polar Team Pro’s distance measurements differed to a greater degree, which weakened validity in the indoor conditions. Notably, Polar Team Pro uses IMU technology as an alternative for indoor conditions due to the lack of satellite signals needed by GPS systems. Although that finding may cast doubt on the applicability of Polar Team Pro devices for indoor use [29], the poor validity for indoor speed measurements when utilizing Polar Team Pro sensors, particularly at higher speeds, limited the ability to precisely quantify external workloads. Consequently, decisions regarding when they can be used may need to be modified.

Playermaker provided good validity in all conditions with a small but significant difference (i.e., <5%) from the actual distance. Although Playermaker measured marginal but significantly lower distances than Polar Team Pro in the straight-line condition, no differences were detected between the two systems for the outdoor rectangle condition. Moreover, Playermaker provided good accuracy for the indoor condition and measured significantly higher distances than Polar Team Pro. Playermaker was not as heavily influenced by changes in conditions (i.e., indoor versus outdoor) as Polar Team Pro was, because the IMUs were not dependent upon a GPS signal. Differences were detected for Playermaker system conditions, and the slight difference between indoor and outdoor conditions could be partly attributed to differences in pitch and ground cover affecting acceleration in the landing phase due to different levels of the hardness of the floor. Although Waldron et al. [25] determined that Playermaker’s devices measured greater total distances than GPS devices, which they attributed to greater distance measurements in speeds below 20 km h^−1^, that outcome was not observed in our study, in which we detected a marginal but significantly longer distance measured with Polar Team Pro in the straight-line condition (refer to Table 2). In the indoor rectangle condition, however, Polar Team Pro measured significantly lower distances than Playermaker.

In the straight-line condition, similarities in speed measurements by Polar Team Pro and actual speeds ranged from a 1.4% overestimation at 8 km h^−1^ to a 2.0% underestimation at 20 km h^−1^. In the same condition and at lower speeds, those results were similar to the outcomes of Playermaker, though larger underestimations were made by Playermaker at higher speeds. Previous studies have shown that GPS devices typically measure below criterion measurements for mean speed and distance, but appear to be consistent [15,16]. Contrary to previous findings [25], Playermaker measured significantly lower speeds for all intensities except for 18 km h^−1^, and significantly lower speed for all intensities with no exceptions in the indoor and outdoor rectangle conditions.

In the outdoor rectangle condition, Polar Team Pro significantly underestimated speed for speeds ranging from 12 to 20 km h^−1^, whereas Playermaker significantly underestimated speeds ranging from 8 to 20 km h^−1^. Both systems demonstrated gradually larger underestimations when speeds increased, which may impair validity when measuring high speeds with either system. The underestimated speeds for Polar Team Pro ranged from 2.5% at 12 km h^−1^ to 9.1% at 20 km h^−1^, whereas underestimated speeds for Playermaker ranged from 1.6% at 10 km h^−1^ to 10.4% at 20 km h^−1^ (see Table 3). Both systems experienced a greater percentage of deviation from prescribed speeds for speeds ranging from 12 to 20 km h^−1^ for the outdoor rectangle condition compared with the straight-line condition. The rectangle condition led to a movement pattern that required turns, accelerations, and decelerations, which may have complicated the capture of speed among the athletes; participants frequently performed 90-degree turns at gradually higher speeds, which may have led to shortening in each corner due to insufficient GPS measurements and a complete lack of measurements in each corner of the rectangle. Furthermore, the anatomical placement of the GPS devices for Polar Team Pro on each participant’s chest may have caused inaccurate measurements in the corners and thereby compromised speed measurements due to the anatomical distance between the device and the athlete’s feet. The Playermaker device, by contrast, was foot-mounted, and therefore not influenced by that circumstance to the same extent [30]. Even though Playermaker was not dependent on satellite signals, it still showed worsening speed measurements during the rectangle conditions. Although speed measurements from the IMUs were determined by measuring accelerations and the angular speed of the athlete and though changes in gait could be detected for each stride to determine speed, it seems to be problematic for the Playermaker system to determine speed when there is significant variety in gait events and rapid changes in stride lengths, stride frequencies, and flight times, as occurred in the rectangle conditions [31].

For the indoor rectangle condition, Polar Team Pro significantly underestimated all speeds ranging from 10 to 20 km h^−1^, with an average percentage difference of 12.7% (see Figure 3). Amid a lack of satellite signals, Polar Team Pro uses IMU technology to measure speed and distance during indoor use, which did not provide satisfactory validity for measuring speed in the rectangle running conditions. Although Playermaker also underestimated speed indoors, speed measurements in the outdoor rectangle condition remained relatively accurate, which is reasonable considering that Playermaker does not depend on satellite signals and is therefore not affected when performing indoors versus outdoors. There was a difference in surfaces, however, with the indoor protocol being performed on a harder surface, and the slightly greater underestimation of speeds in the indoor protocol than in outdoor conditions could therefore have resulted from different mechanical impulses from the accelerometer in the IMU that provide different bases to determine speed.

Both systems underestimated distance and speed during the straight-line condition but appeared to be consistent and provide good reliability (see Table 3 and Figure 3). On average, the random error between the Playermaker and Polar Team Pro sensors was less than 2% for both systems in that condition, which indicates good reliability between the different sensors when measuring distance and speed during straight-line running (see Table 5 and Figure 4). Test–retest reliability varied with ICCs ranging from 0.471 (i.e., low) to 0.994 (i.e., very high), and significantly lower ICCs were found at 8, 10, and 12 km h^−1^ than with the other three speeds (see Table 4); the low ICCs at those low speeds could be due to small changes in the measured distances between the runs (±1–2 m), which already had a noticeable effect when calculating the ICCs. The small differences in measured distance also resulted in lower ICCs for Polar Team Pro in the straight-line condition than for Playermaker, because Polar Team Pro’s measurements were very accurate at those low speeds (see Table 3). Polar Team Pro’s highly accurate measurements in the straight-line condition also resulted in higher but still accurate CVs. By contrast, the CV test–retest measurements were accurate under all conditions, which indicates that although the individual units for both systems are reliable, the comparison of the units and measuring systems should be avoided when measuring indoors.

Our study reveals a systematic error resulting in consistently underestimated total distances for both systems and under each protocol. Measurements in outdoor conditions were stable and reliable, and the values of measured distances were in line with the values in other studies investigating those systems [15,25]. The relatively low number of participants may be a limitation of our study, despite the simultaneous use of several devices to increase the sample. Future research should include measurements from both systems in complex situations with varying speeds in order to assess the systems’ validity and reliability in conditions that typically occur in team sports.

## 5. Conclusions

Polar Team Pro and Playermaker significantly underestimate distance in indoor and outdoor conditions. Both systems measured close to the actual distances (i.e., <5%) for the straight-line and rectangle outdoor conditions, thereby suggesting the relatively low significance of distance measurements in the sports context in which the systems are applied. In the indoor condition, only Playermaker measured close to the actual distances (i.e., <5%). The validity also varied to a greater degree when speeds increased. In the straight-line condition, Polar Team Pro overestimated speed at 8 km h^−1^ and underestimated speed at 20 km h^−1^. When measuring all other speeds, Polar Team Pro provided good validity in the straight-line condition and when measuring speeds in the rectangle condition of 8 and 10 km h^−1^, while the system underestimated average speed as speeds increased. Meanwhile, Playermaker underestimated average speed in all conditions and at all speeds. Last, Polar Team Pro provided poor validity and reliability in indoor conditions.

## Figures and Tables

**Figure 1 sensors-23-08251-f001:**
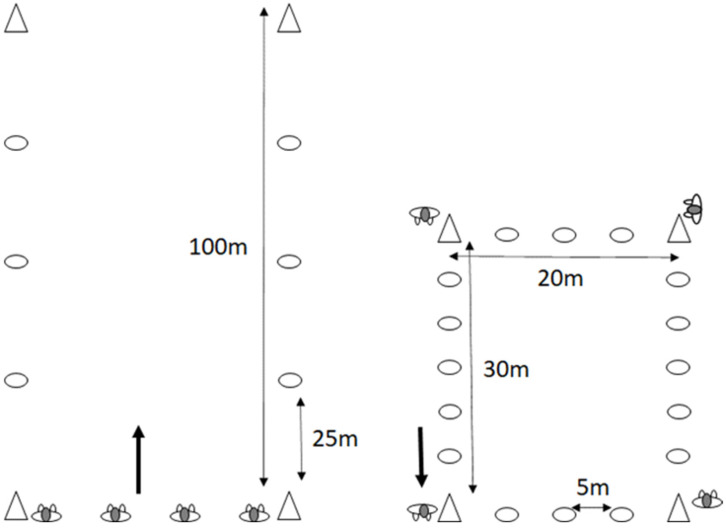
Setup of the different running conditions (i.e., straight-line and rectangle).

**Figure 2 sensors-23-08251-f002:**
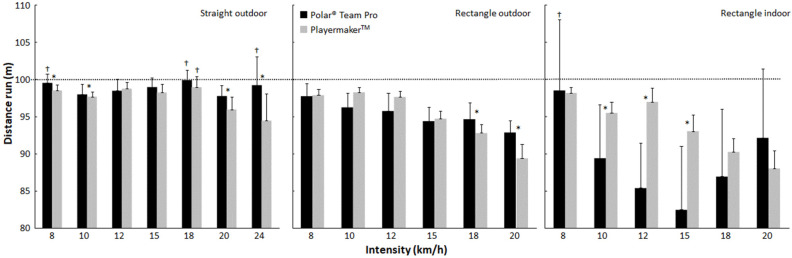
Measured mean (±SD) distance per speed and condition (i.e., outdoor straight line, outdoor rectangle, and indoor rectangle) with Polar Team Pro and Playermaker for a prescribed distance (dotted line). * Significant difference in distance between the two measuring devices at the same speed for the same condition (*p* < 0.05). ^†^ No significant difference in distance for the prescribed distance for the condition and speed with the same measuring device (*p* < 0.05).

**Figure 3 sensors-23-08251-f003:**
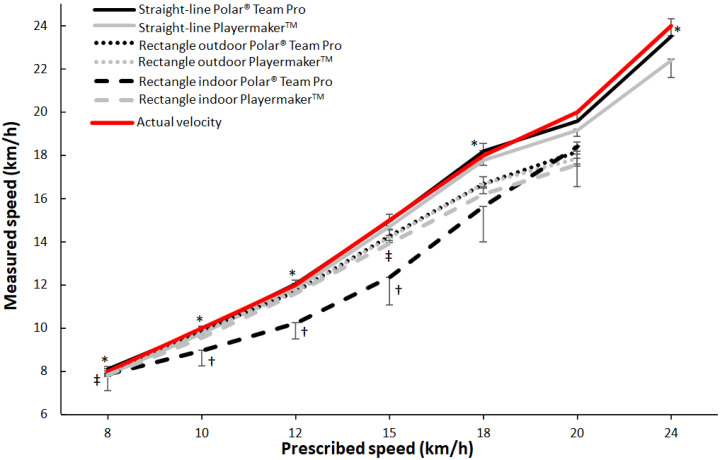
Calculated mean speed (±SD) and conditions for Polar Team Pro and Playermaker. * Significant difference in calculated speed between the two measuring devices at the speed for the straight-line condition (*p* < 0.05). ^†^ Significant difference in calculated speed between the two measuring devices at the speed for the rectangle outdoor condition (*p* < 0.05). ^‡^ Significant difference in calculated speed between the two measuring devices at the speed for the rectangle indoor condition (*p* < 0.05).

**Figure 4 sensors-23-08251-f004:**
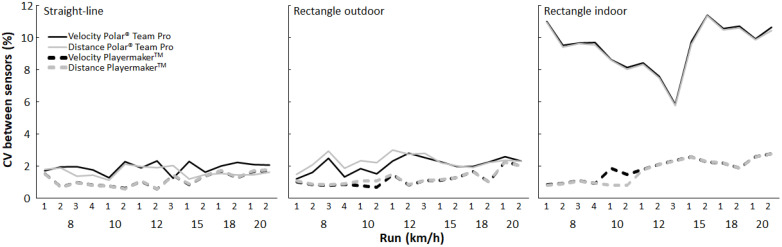
Coefficient of variation (CV) between Polar Team Pro and Playermaker sensors for every run and condition.

**Table 1 sensors-23-08251-t001:** Running protocol with different distances, intensities, and periods of rest between runs.

Running Speed (km h^−1^)	8	10	12	15	18	20	24
Straight line							
Distances (m)	4 × 100	2 × 100	3 × 100	2 × 100	2 × 100	2 × 100	2 × 100
Rest between runs (s)	0	0 (48 s after second run)	0 (30 s after third run)	36	40	42	45
Total time (min) accumulated	3	5	7	9	11	13	15
Outdoor rectangle							
Distances (m)	4 × 100	2 × 100	3 × 100	2 × 100	2 × 100	2 × 100	
Rest between runs (s)	0	0 (48 s after second run)	0 (30 s after third run)	36	40	42	
Total time (min) accumulated	3	5	7	9	11	13	
Indoor rectangle							
Distances (m)	4 × 100	2 × 100	3 × 100	2 × 100	2 × 100	2 × 100	
Rest between runs (s)	0	0 (48 s after second run)	0 (30 s after third run)	36	40	42	
Total time (min) accumulated	3	5	7	9	11	13	

**Table 2 sensors-23-08251-t002:** Mean (±SD) total distance and percentage difference with actual distance measured for each condition with Polar Team Pro and Playermaker.

		Actual Distance	Polar Team Pro	Playermaker
Straight line	Distance (m)	1700	1680 ± 21 ^†^	1660 ± 16
	Difference (%)		−1.1 ± 1.2 *	−2.3 ± 1.0 *
Outdoor rectangle	Distance (m)	1500	1423 ± 26	1443 ± 11
	Difference (%)		−4.5 ± 1.6 *	−3.3 ± 0.7 *
Indoor rectangle	Distance (m)	1500	1351 ± 111 ^†^	1417 ± 22
	Difference (%)		−8.7 ± 6.5 *	−4.9 ± 1.3 *

All measured distances in all conditions were significantly lower than actual distance. ^†^ Significant difference compared with the other measuring device (*p* < 0.05). * Significant difference compared with the other conditions but using the same device (*p* < 0.05).

**Table 3 sensors-23-08251-t003:** Percentage difference (i.e., negative values indicate overestimation) for Polar Team Pro and Playermaker for prescribed speeds for each condition.

Speed	8	10	12	15	18	20	24
Straight-line							
Polar Team Pro	−1.4 *	0.8	−0.5	0.1	−1.2	2.0 *	2.0
Playermaker	1.7 *	2.2 *	1.3 *	1.8 *	1.1	4.0*	6.5 *
Outdoor rectangle							
Polar Team Pro	−0.3	0.9	2.5 *	4.9 *	7.3 *	9.1 *	
Playermaker ^†^	2.1	1.6	2.4	5.3	7.3	10.4	
Indoor rectangle							
Polar Team Pro	1.6	10.4 *	14.7 *	17.6 *	13.1 *	7.9 *	
Playermaker ^†^	1.9	4.4	3.1	7.1	9.8	12.0	

* Significant difference with prescribed speed at the speed (*p* < 0.05). ^†^ Significant difference with prescribed speed at all intensities (*p* < 0.05).

**Table 4 sensors-23-08251-t004:** Intraclass correlations (ICCs) per described speed condition for speed and distance with Polar Team Pro and Playermaker.

Speed	8	10	12	15	18	20	24
Straight line							
Polar Team Pro	0.471	0.717	0.421	0.871	0.760	0.950	0.899
Playermaker *	0.661	0.915	0.653	0.911	0.943	0.989	0.994
Outdoor rectangle							
Polar Team Pro	0.592	0.812	0.807	0.832	0.919	0.899	
Playermaker	0.986	0.882	0.458	0.858	0.796	0.944	
Indoor rectangle							
Polar Team Pro ^†^	0.986	0.939	0.973	0.973	0.986	0.978	
Playermaker	0.925	0.804	0.957	0.972	0.955	0.987	
Distance							
Straight line							
Polar Team Pro	0.750	0.689	0.772	0.913	0.896	0.916	0.946
Playermaker	0.660	0.892	0.654	0.923	0.950	0.986	0.994
Outdoor rectangle							
Polar Team Pro	0.873	0.777	0.87	0.867	0.829	0.903	
Playermaker	0.899	0.868	0.49	0.858	0.812	0.953	
Indoor rectangle							
Polar Team Pro ^†^	0.985	0.938	0.972	0.971	0.985	0.981	
Playermaker	0.927	0.805	0.960	0.974	0.953	0.987	

* The ICC for Playermaker with Polar Team Pro was significantly higher in the condition. ^†^ The ICC for the condition with Polar Team Pro was significantly higher than in the other two conditions measured with Polar Team Pro.

**Table 5 sensors-23-08251-t005:** Coefficients of variation (CVs) per prescribed speed for speed and distance with Polar Team Pro and Playermaker.

Speed	8	10	12	15	18	20	24
Straight line							
Polar Team Pro *	1.48	5.64	1.57	1.09	1.06	0.61	1.23
Playermaker	0.85	3.04	1.62	0.77	0.41	0.46	0.36
Outdoor rectangle							
Polar Team Pro	2.75	1.06	1.80	0.86	1.00	1.67	
Playermaker	1.58	0.29	1.73	0.65	1.41	1.84	
Indoor rectangle							
Polar Team Pro *	2.33	2.49	2.37	2.48	2.07	1.64	
Playermaker	1.33	2.56	1.49	2.30	1.25	0.69	
Distance							
Straight line							
Polar Team Pro *	1.23	6.39	2.85	0.48	0.64	0.79	1.09
Playermaker	0.86	3.10	1.69	0.80	0.38	0.42	0.37
Outdoor rectangle							
Polar Team Pro	2.17	1.08	1.38	0.91	1.27	2.01	
Playermaker	1.57	0.33	1.78	0.64	1.42	1.80	
Indoor rectangle							
Polar Team Pro *	2.36	2.46	2.36	2.55	2.05	1.61	
Playermaker	1.32	2.51	1.47	2.28	1.22	0.71	

* The CV was significantly higher for Polar Team Pro than Playermaker for the condition.

## Data Availability

The data used in the study are available upon request from the corresponding author. The data are not publicly available due to Norwegian laws concerning privacy.

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
