# Peer review of "Validity and Reliability of Polar Team Pro and Playermaker for Estimating Running Distance and Speed in Indoor and Outdoor Conditions"

_sensors, 2023, doi:10.3390/s23198251_

Round 1
Reviewer 1 Report (New Reviewer)
Summary:
Authors compared two regularly used devices in team sports which measure athletes speed and distance run usually used for training sections namely Polar Team Pro and Playmaker. To this end, they set up outdoor and indoor running courses (100 meters straight line and 100 meters rectangular form) and had four participants to complete these courses at different prescribed speeds ranging for 8 km/h to 24 km/h. Each distance and/or course was completed at least twice with appropriate resting times between runs. Following the tests, the measured total running distance and average speed were compared between the devices and different courses. Authors used appropriate and well selected statistical methods to evaluate the differences and came to acceptable conclusion. The structure of the paper is logical and follows the expectations of the Journal. The figures and tables are informative and the structure is clear. As these devices are commonly used in team sports their validation is important and useful for the sports community. The main finding of the measurements is that both devices underestimate both the distances covered and the speed taken. This observation was true for both indoor and outdoor conditions. Furthermore, the reliability of the devices appeared acceptable except for Polar Team Pro indoor.
Results:
· The results are well documented however certain clarifications need to be done. Considering Figure 2 it is clear that the large SD doesn’t allow the comparison of the same device at different speeds for the rectangular indoor course. See e.g. 15 km/h vs 8 km/h or 15 km/h vs 20 km/h for Polar Team Pro. Authors should have repeated the measurements to reduces SD and thus enable such comparisons. It is intriguing that the underestimation of the distance at 15 km/h is greater than either at 8 km/h or 20 km/h. Is this real in Authors’ opinion?
· It would have been interesting to see how changing the devices between participants affects the results, if any. In other words, would the individually possibly different movement pattern influence the results?
· Line 235: What does F14, 154 signify?
· Table 4, First row: What might be the reason for the low value at 12 km/h as both the values at 10 and 15 km/h are much higher? Similar question stands for the value at 12 km/h in line 4.
Discussion:
‐ Line 313: typo: appear instead of “appears”.
Quality of English Language:
‐ Fine, no issues detected.
Author Response
Please see the attachment

Reviewer 2 Report (New Reviewer)
1. A small number of subjects may lead to bias in results. so this paper need more subjects to test and then the result can be convincing.
2. Whether the different materials of indoor and outdoor sites will have an impact to the results.
3. More sets of data should be measured when the four corners of the rectangular condition change direction.
4. Whether the waistband in front of the chest has a large error in testing the turn.
5. The influence of GPS signal strength should be considered to the test.
Some English expressions should avoid the common expressions in Chinese, and English expressions should be adopted.
Author Response
Please see the attachment

This manuscript is a resubmission of an earlier submission. The following is a list of the peer review reports and author responses from that submission.
Round 1
Reviewer 2 Report
Thank you for giving me an opportunity to review the manuscript titled “Validity and reliability of the Polar Team Pro and Playermaker systems for the estimation of running distance and speed in in-door and outdoor conditions”. Please find below my comments and suggestions for improvement.
Abstract: The first sentence needs to be rewritten (e.g., …that will take into account that both… does not make sense). I would suggest authors separate this introductory sentence in two different sentences.
Abstract: The purpose of the present study was… (past tense). Also please reconfigure, age=30.0±5.1 years, body mass=76.3±5.3 kg, height=1.79 ± 0.09 m.
Abstract: What was your gold-standard measurement (criterion measure)? “Both systems underestimated the total distance”… How? Compared to what?
Abstract: ICC for what? Absolute agreement or consistency of the measurements? This needs to be specified.
Line 34: This sentence needs to be rewritten. I understand what you are trying to say here, but it is not properly stated. For example, “monitoring external loads that athletes are exposed to during regular training sessions is …”.
Line 36: “beneficial recovery phase…” does not make sense. I agree with the idea, but this sentence needs to be reformatted. I would also encourage authors to check a recently published book by Zatsiorsky, Kraemer, and Fry (2020).
Line 40: “in many sports…” Provide examples (e.g., …).
Line 41-47. This section needs to be rewritten. In team sports like soccer, however… is not proper. The same comment applies to “…speed during training, making it more difficult to control speed during training sessions…”.
Line 52-54: Please check grammar here.
Line 57: Please define “short sprints” - 10m, 15m, 30m?
Line 64: You state “studies” as plural, but only one reference is listed at the end of this sentence.
Line 82-88: This sentence is way too long. This needs to be simplified. I understand the point you are trying to make here, but this sentence needs improvement.
Line 90: Wouldn’t professional sports teams actually be the ones that actually have a budget to purchase this technology (e.g., NBA, NFL, Premier League, Euro League)? Wouldn’t it be more appropriate to say for high school, collegiate, junior, or amateur teams?
Line 93: You cannot end the sentence with “however”. The same applies to “therefore” in the middle of the sentence. So, I will abstain from making further comments pertaining to the grammatical errors as this needs to be considerably improved throughout the overall manuscript.
Line 103: “…after total distance, distance…”. Please correct.
Line 108: Four participants? Did you do a power analysis? This is very low, even for the sports science field of research.
Line 113: What was warm-up composed of? Running or jogging? Dynamic stretching exercises?
Line 114: Why testing procedures were conducted in a group setting rather than individually? Also, what system was used to give audible signals?
Line 132: How much experience? How was this determined?
Line 168: The same comment applies to ICC as in the abstract section.
Line 179: Did you perform the Shapiro-Wilk test?
Table 2: Significant difference when compared to other measurement equipment. Specify this. Also, you do not need this statement in the table (“…all measured distances in all conditions were significantly lower than the actual distance…”), this can be seen by the magnitude of the values presented in the table.
Line 183: If you listed F values, please list degrees of freedom.
Line 260: Please correct this reference to Huggins et al. [5]
Figure 4: “Run” – you mean velocity in km/h? Also, define what a CV is in the Figure 4 title. The figures in the manuscript should be observed as independent of the text. Why is there a large difference in CV between outdoor and indoor conditions?
Line 252: Can these conclusions actually be made only on four participants? In my opinion, this is a large limitation of the present investigation.
Line 257: 1.1% should be emphasized as not being of “practical significance” despite reaching the level of statistical significance. This is not large in a practical setting if we consider that reliability levels are solid.
Line 270: Why is reference 29 listed here? Fox et al.? What the authors are trying to accomplish by stating a similarly conducted project here.
Line 295: Please rephrase this sentence and pay attention to the grammar.
Line 316: I would encourage authors to check the recently published paper by Cabarkapa et al. (2023) in the MDPI Sports journal “Impact of the anatomical accelerometer placement on vertical jump performance characteristics”. This may be used to support the claims that you made in this paragraph.
Line 356: You need to expand on the limitations of the present study, especially since you only had four participants.
Line 365: Buy how much (overestimation or underestimation)? Please be specific as this is important to state. Also, you should emphasize the practical aspect of these findings. For example, 1.1% may not be significant in the applied sport setting.
Line 371: You need to list your recommendation and conclusion. What do these findings tell practitioners? What system should they use based on these findings?
References: Please check reference formatting.
The quality of the English language needs to be notably improved in my opinion.
Round 2
Reviewer 1 Report
None of my doubts in this research have been satisfactorily answered. It is said to be obtained from 100Mm raw data. How was it determined that it was the 100th meter in the raw data. I'm very curious about this. The sample size in the study with 4 people is really low. These problems are large and research is sloppy. I invite researchers to redesign the research. Sorry, this research cannot be published.
Reviewer 2 Report
Thank you for incorporating my suggestions for improvement.
Moderate editing of English language is still required.